

# Tracing truth: dynamic temporal networks for multi-modal fake news detection

Jiaen Hu, Juan Zhang and Zichen Li

College of Information Engineering, Beijing Institute of Graphic Communication, Beijing, China

## ABSTRACT

As the internet continues to evolve rapidly and social media becomes increasingly prevalent, the ways people access information has become increasingly diverse. However, the proliferation of fake news has emerged as a critical problem, presenting major challenges to the integrity of the information ecosystem. To address the complex propagation mechanisms of fake news, existing studies leverage multi-modal information and dynamic propagation social graphs for effective detection. Nonetheless, capturing the temporal relationships of propagation nodes in dynamic social networks accurately and dynamically integrating multi-modal information for improved detection accuracy remains a technical challenge. In response, This study proposes a multimodal approach to fake news detection—the dynamic temporal network (DTN) model. Firstly, this model designs a time similarity strength metric to measure the temporal similarity among nodes in propagation sequences and introduces a weighting mechanism to dynamically fuse multi-modal information. Secondly, it constructs a social propagation graph model, enhancing node representation through the dynamic variations of time similarity and graph structure, and utilizes the Transformer encoder to extract the overall semantic features of news propagation. Furthermore, the model views the news propagation process as a complex system, analyzing the temporal dynamics of news in real social networks, effectively revealing the abnormal propagation patterns of fake news. Further analysis demonstrates that the proposed DTN model exhibits high accuracy and effectiveness in multi-modal fake news detection.

## INTRODUCTION

With the swift advancement of the internet and the widespread growth of social media, the ways in which people access information have become increasingly diverse. We no longer rely on printed materials; instead, we communicate with the outside world through quicker and more comprehensive social media channels. However, alongside the convenience that social media brings to our lives, hidden drawbacks are gradually eroding our daily experiences. One significant issue is fake news. While internet platforms facilitate the rapid flow of information, they also inadvertently provide a breeding ground for the spread of falsehoods. Individuals in various sectors spread fake news either to gain attention and small profits or, in more severe cases, to intentionally incite social unrest,

Corresponding author
Juan Zhang, zhangjuan@bigc.edu.cn

potentially harming national interests. For instance, during the 2020 COVID-19 outbreak, false claims that 5G radiation could spread the virus propogated on social media, causing panic. This led to 5G towers being damaged or burned in the UK and the Netherlands, disrupting communication systems (*Schraer & Lawrie, 2020*). Consequently, the issue of fake news has become a focal challenge in the realm of cybersecurity, attracting widespread interdisciplinary research. Scholars from fields ranging from complex network analysis to communication theory, sociology, psychology, and artificial intelligence are actively exploring the intrinsic mechanisms of fake news propagation and working to develop more accurate detection technologies. Their goal is to provide the public with effective tools for identifying fake news, thereby helping to maintain the purity and health of the information ecosystem. Hence, it is imperative to develop a highly efficient and precise model for detecting fake news.

To detect fake news, classic text-based methods (*Cheng, Nazarian & Bogdan, 2020*) use variational autoencoders (VAEs) to encode textual information, generating embedded representations of news texts and improving performance through multi-task learning. In social networks, connections between news and entities such as users and comments make graph-based methods effective. *Yin et al. (2024)* introduced self-supervised learning with a graph autoencoder, while *Bian et al. (2020)* utilized graph convolutional networks (GCNs) with directed rumor graphs to learn propagation and dispersion patterns. News content includes text, images, videos, audio, and more. Researchers have proposed leveraging multimodal information to improve fake news detection, enabling models to better understand news content for more accurate results. *Xue et al. (2021)* emphasized the consistency of multimodal data, capturing the overall characteristics of social media information, while *Yadav & Gupta (2024)* leveraged emotional cues and a vision transformer to filter irrelevant data and boost classification performance.

While current methods for detecting fake news have shown some effectiveness, they still exhibit certain limitations, primarily reflected in the following aspects:

1. **Limitations of graph structure representation** Many studies use news text as the sole information source for graph nodes, pooling them into graphs or subgraphs for classification. This approach fails to capture complex node-edge relationships and dependencies in news propagation, limiting its ability to represent the propagation network's features.

2. **Lack of dynamic information fusion** The majority of models center on the content of news, neglecting both the social environment and its dynamic spread. As public understanding of news content deepens, propagation paths evolve, making real-time monitoring crucial for detecting fake news. However, the lack of effective integration of dynamic information limits the models' performance in complex environments.

To tackle these challenges, this study introduces a dynamic temporal network (DTN) model, designed to address the complexities of multimodal fake news detection. As fake news continues to spread rapidly across social media platforms, its multi-modal characteristics and dynamic propagation patterns make detection increasingly complicated. Following this line of thought, we designed the DTN model, which enhances detection

capabilities by exploring the complex temporal relationships between nodes, calculating the temporal and spatial distribution features during news propagation, and capturing the dynamic changes in graph structures. Based on this, we address three core challenges: (1) How to effectively capture the temporal dynamics of nodes? (2) How to dynamically integrate multi-modal information to maximize the complementary aspects during the propagation process? (3) How to reveal the abnormal patterns of fake news with respect to temporal and spatial distribution to improve detection accuracy?

To address these challenges, *Yu et al. (2017)* introduced temporal similarity metrics to link prediction, showing that nodes frequently infected at similar times are more likely to connect. Building on this, we designed a temporal similarity strength metric to dynamically weight neighboring nodes in propagation sequences, capturing temporal dynamics. We also constructed a propagation social graph model combining temporal similarity with dynamic graph changes, enhancing node representation and capturing local contextual features. Inspired by *Sheng et al. (2022)*, who noted that fake news aligns with popular events to maximize exposure, we introduced entropy analysis to quantify temporal dynamics, revealing abnormal patterns in fake news propagation. To improve global semantic perception, we used a Transformer encoder to capture global semantics and integrate multi-modal information, significantly enhancing detection accuracy. Experiments show that the DTN model outperforms existing methods in accuracy and robustness, effectively capturing propagation dynamics, integrating multi-modal information, and detecting fake news efficiently.

The key features and advancements of our model can be summarized as:

- **Integrality** We proposed a time similarity strength metric and a dynamic weighting mechanism for the integration of multi-modal information among nodes, improving the model's capacity to understand semantics throughout the propagation process.
- **Efficiency** We constructed the DTN model, which combines graph structure with the time similarity metric and utilizes a Transformer encoder to capture global semantics, thereby improving the effectiveness of fake news detection.
- **Monitor** Through feature analysis, we revealed the complexities of the concentrated short-term propagation and long-term diffusion of fake news, identifying patterns in their temporal and spatial distributions, thus enabling early monitoring of news propagation and long-term diffusion warning functions.

Experimental results validate the DTN model's significant detection performance and generalization capability across different datasets, demonstrating its potential in capturing news propagation dynamics and identifying fake news.

## RELATED WORK

### Text-based methods

Traditional fake news detection methods focus on analyzing textual content by extracting semantic features (*Madani, Motameni & Roshani, 2024*). *Yu et al. (2017)* first applied CNNs to model news articles, mapping related posts into vectors, concatenating them into a matrix, and extracting features with CNNs before classification. *Cheng, Nazarian*

*& Bogdan (2020)* employed a variational autoencoder (VAE) for encoding news text, generating embeddings and enhancing performance through multi-task learning. *Vaibhav, Mandyam & Hovy (2019)* represented news using a graph structure, where sentences acted as nodes and their similarities formed edges, utilizing GCNs to combine node information and identify fake news.

However, these methods rely solely on text, neglecting user behavior and social data in social networks. This limits their ability to capture fake news dissemination characteristics, as user interactions and propagation patterns provide a more comprehensive basis for detection.

## Graph-based methods

In social networks, connections between news and entities like users and comments can be utilized for fake news detection by constructing homogeneous or heterogeneous graphs (*Ramya & Eswari, 2024*; *Jiang et al., 2024*; *Su et al., 2024*). *Dou et al. (2021)* evaluated user credibility by considering posting history as an internal element and news propagation as an external aspect. *Shu, Wang & Liu (2019)* modeled relationships like publisher-post-news and user-spread-news in a heterogeneous information network, using matrix factorization to enhance node representations and detection accuracy. *Park & Chai (2023)* integrates user, content, and social network features based on social capital, effectively reflecting fake news propagation characteristics.

However, these methods often fail to capture complex node-edge relationships and dependencies in news dissemination processes.

## Multi-modal methods

News content includes text, images, videos, and audio. Researchers have proposed multimodal approaches to improve fake news detection (*Zhang et al., 2024*; *Zhang et al., 2025*; *Zhu et al., 2024*) proposed a reinforcement-driven subgraph selection method, adaptively retrieving entity-level knowledge and capturing cross-modal correlations *via* heterogeneous graph learning. *Luvembe et al. (2024)* introduced complementary attention fusion between image captions and text, combined with feature normalization to reduce semantic noise and improve detection performance. *Peng et al. (2024a)* emphasized that fake news is not always semantically similar, and proposed contextual semantic learning to fuse global and local semantics for more robust detection in multimodal environments.

However, these models lack effective integration of dynamic information. Real-time monitoring of news dissemination is crucial, as public understanding evolves over time, affecting fake news spread. Dynamic information integration is essential for addressing real-world complexities.

To tackle these issues, we introduce the DTN framework, designed to surpass the constraints of text-centered methods. It captures complex graph relationships and node dependencies during news dissemination while enhancing semantic information. By analyzing spatiotemporal distribution patterns, DTN dynamically weights nodes for comprehensive multimodal fusion, excelling in distinguishing real and fake news on large social networks.

# PROPOSED MODEL

To address the limitations of graph structure representation and the lack of dynamic information fusion, this paper proposes the DTN model, designed to effectively identify the authenticity of news. This section introduces the design and implementation of DTN. Figure 1 illustrates the architecture of DTN, which consists of five modules: feature representation, graph structure enhancement, temporal feature analysis, temporal dynamics fusion, and global semantic encoding.

## Feature representation module

Figure 2 outlines the architecture of the feature representation module, which is responsible for extracting multi-modal information from news events and their social context. This module incorporates textual, visual, and social media signals, enabling the model to capture both semantic content and user interaction dynamics. By performing quantification and normalization across heterogeneous modalities, the model ensures feature alignment and dimensional consistency, laying a solid foundation for downstream tasks. The module consists of three components: (1) text feature representation, which encodes the semantics of news titles, content, and social posts; (2) image feature representation, which captures visual cues from accompanying news images; and (3) social media feature representation, which models user attributes and engagement metrics. Together, these representations provide a comprehensive understanding of event credibility and significantly enhance the model's predictive performance.

### *Text feature representation*

To obtain meaningful vector representations for diverse textual modalities, we employ a hybrid encoding strategy leveraging two pre-trained language models. Given an input sentence $S$, we first tokenize it with padding or truncation into a fixed-length sequence $T = \{[CLS], s_1, s_2, \ldots, s_i \ldots, s_n, [sep]\}$, where $s_i$ denotes the token corresponding to the $i$-th word. This sequence is passed into a fine-tuned robustly optimized bidirectional encoder representations from Transformers (RoBERTa) encoder to extract contextualized embeddings. The hidden state of the special token $[CLS]$ is used as the sentence-level embedding, denoted by $e \in \mathbb{R}^{d_t}$, where $d_t$ is the embedding dimension. In parallel, we employ a fine-tuned T5 encoder to process structured textual fields in the news data, including the title, content, and associated post. Each field is encoded independently to produce dense vector representations $e^t$, $e^c$, and $e^p$, respectively. These embeddings are further normalized to ensure consistency across modalities and facilitate downstream training. The RoBERTa-based representation is specifically used for user-related textual inputs, such as user descriptions or profile metadata, yielding the user embedding $e^u$. This dual-model setup enables the system to capture both general linguistic patterns and field-specific semantics across heterogeneous textual inputs.

### *Image feature representation*

To incorporate visual modality into the model, we utilize a pre-trained ResNet18 network to extract semantic-level image features from news-related visual content. Given an input

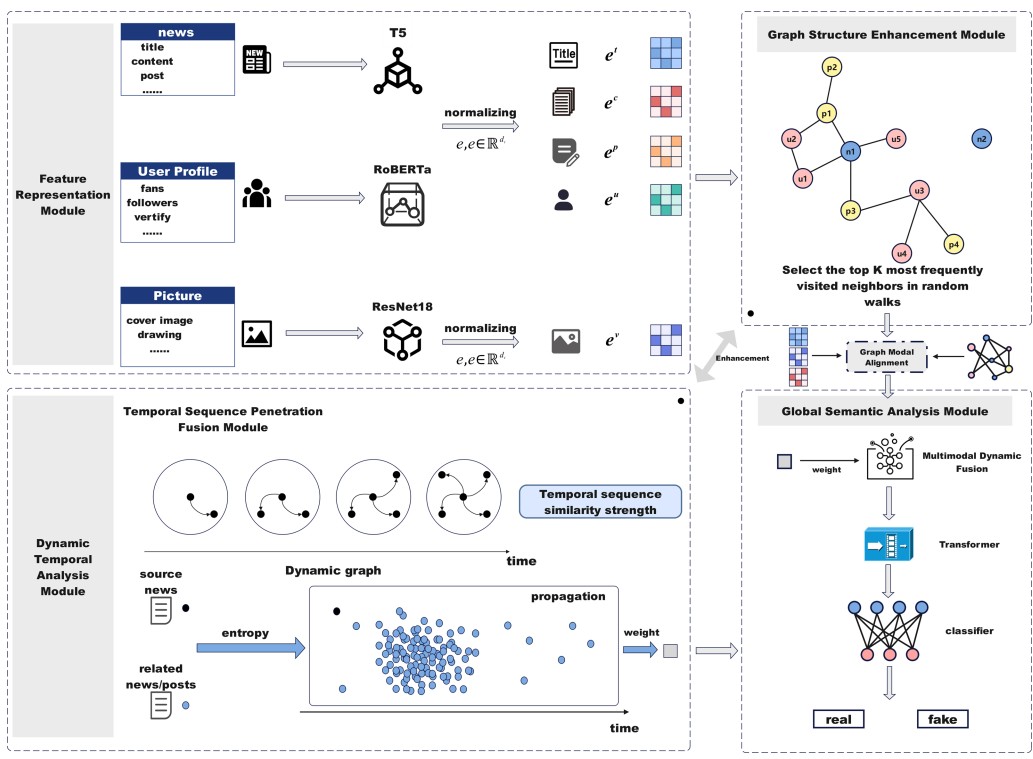

**Figure 1  The framework of the DTN model.**

image associated with a news event, we pass it through ResNet18 and extract the feature vector from its final average pooling layer, resulting in a dense representation. This visual feature vector is then subjected to normalization to ensure alignment with the dimensions of other modalities. The resulting image embedding is denoted as $e^v \in \mathbb{R}^{d_v}$, where $d_v$ indicates the dimension of the image feature vector. By integrating visual cues from news images, the model is able to capture multimodal signals that may reflect emotional tone, contextual clues, or visual bias, thereby enhancing its capacity for comprehensive news understanding and veracity assessment.

### Social media data feature representation

When a news topic emerges, it inevitably triggers public opinion through social media, and metrics such as the number of reposts and likes on posts related to the topic are key numerical attributes that need to be considered. In particular, we need to focus on the social media data of the user $u_i$ who posted the news and related posts. This data contains key attributes, including follower count, fan count, and the verification status of the user. These user features should be treated as important numerical attributes and further incorporated into the model for analysis. Subsequently, the aforementioned social media numerical features are converted into a sparse feature matrix using one-hot encoding, with the embedding representation of the user's social media metrics denoted as $e^u$. In this way, the numerical attributes can be normalized. Quantifying these social media data not

*Peer*J Computer Science

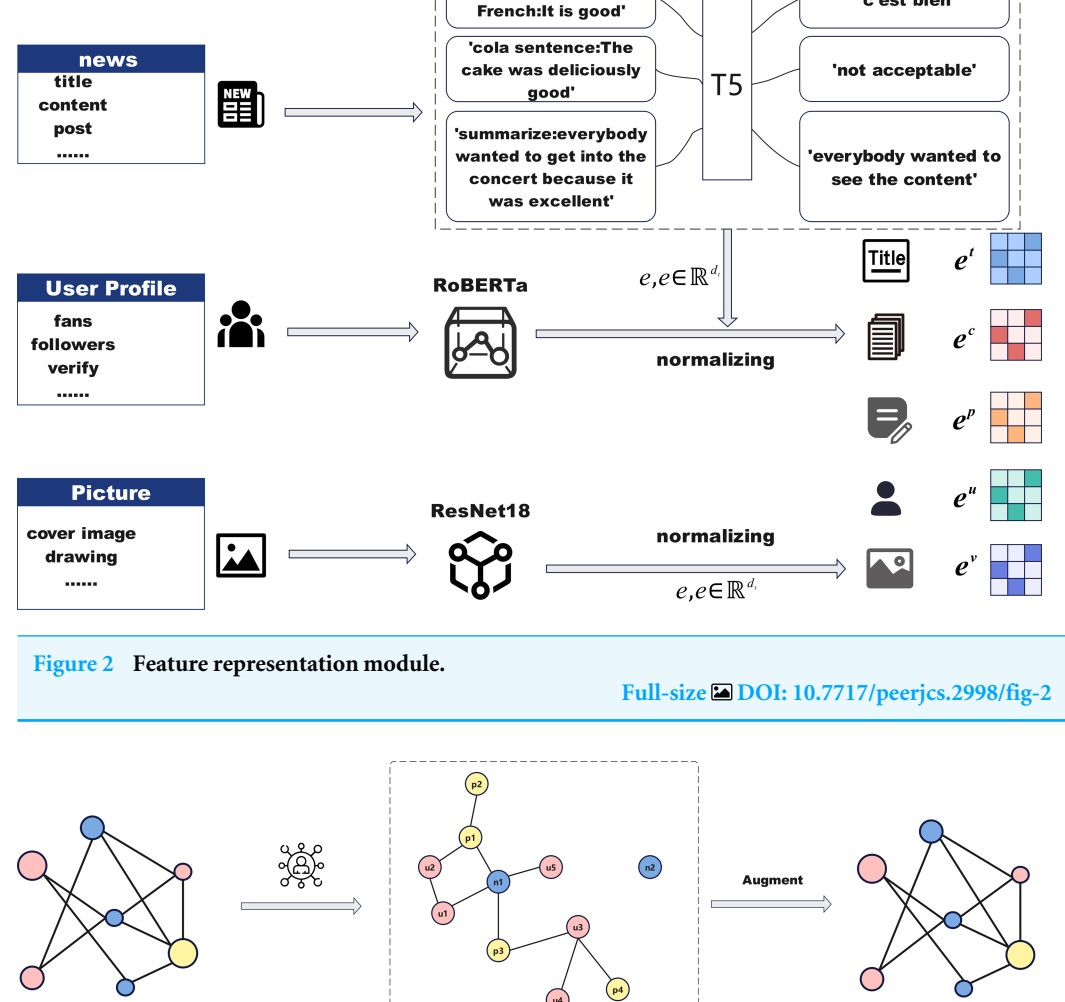

**Figure 2** Feature representation module.

**Figure 3** Graph structure enhancement module.

only helps to fully understand the user's influence and credibility on the platform but also provides more accurate and comprehensive information support for the model, improving its prediction capability and effectiveness.

## Graph structure enhancement module

Relying solely on news text to determine authenticity has limitations, as relationships between users, news, and posts influence information dissemination and public perception. We extend social media data into a multidimensional graph structure to explore the impact of these relationships on fake news detection. Figure 3 illustrates the graph structure enhancement module. By constructing a heterogeneous graph consisting of users, news,

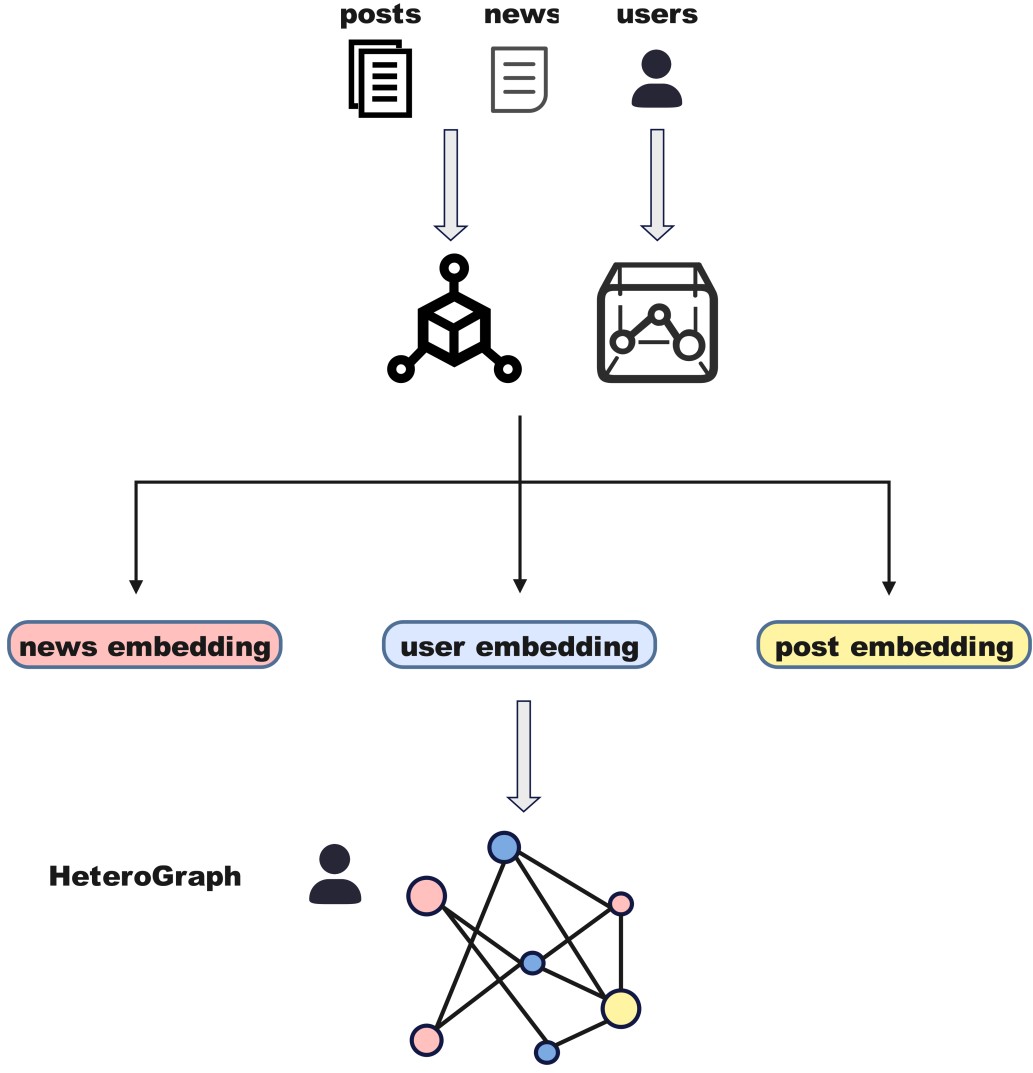

**Figure 4** Construction of heterogeneous graph.

and posts, and optimizing it using random walks and frequency sampling, we capture dynamic propagation relationships and preserve key connections.

### Definition of graph structure

As shown in Fig. 4, we model the relationships between news using a graph structure to further analyze and detect fake news propagation patterns. Drawing from the entity categories identified in the dataset, including users, news, and posts—we construct a heterogeneous news graph $\mathbb{G} = \{V, \epsilon\}$, where $V = \{U, N, P\}$ represents the set of nodes for users, news, and posts, respectively. $U = \{u_1, u_2, \ldots, u_i \ldots, u_{c_u}\}, N = \{n_1, n_2, \ldots, n_i \ldots, n_{c_n}\}, P = \{p_1, p_2, \ldots, p_i \ldots, p_{c_p}\}$ represent the sets of users, news, and posts, with $c_u, c_n, c_p$ denoting the number of users, news, and posts, respectively. $\epsilon$ represents the edges that link these nodes. Specifically, we model news, users, and posts as nodes, with

different types of interactions as edges, constructing a multi-type edge graph that reflects the propagation patterns in social networks. The implementation steps are outlined below:

**Nodes:** The graph nodes correspond to three categories of entities:

(a) **User nodes:** Each user posting news is represented as a node in the graph.

(b) **News nodes:** Each news is represented as a node.

(c) **Post nodes:** Each post related to the news is represented as a node.

**Edges:** Based on different types of interactions between news, we define the following edges between nodes:

(a) **news-post interaction edges:** Recursively add $n - p$ edges between main news nodes and their directly replied posts. Add $p - p$ edges between reply posts to represent interactions among posts.

(b) **user interaction edges:** Add $n - u$ or $p - u$ edges between each main news or post and its corresponding user. If a post has replies, add $u - u$ edges to represent interactions between users.

### Heterogeneous graph optimization

After constructing a network graph based on different node types (users, news, and posts), as shown in Fig. 3, a random walk process is simulated between nodes to capture the underlying patterns of information propagation, going beyond the relationships of first-order neighbors to obtain a more comprehensive propagation graph. The specific steps are as follows:

  i. **Random walk initiation:** Start from an initial node and iteratively visit its neighboring nodes according to the connections defined in the adjacency list.

 ii. **Frequency sampling to select high-frequency neighbors:** In large-scale networks, connections between nodes are often dense. To reduce complexity, a frequency sampling method is used to retain only high-frequency neighbors, controlled by parameter $k$. Neighbor nodes are ranked by visitation frequency during random walks, excluding the starting node and irrelevant nodes. High-frequency nodes are prioritized and retained in the final neighbor list.

iii. **Enforced retention of important edges:** To ensure that certain critical edge types (*e.g.*, news-user, post-user) are always retained, these edges are forcibly added to the neighbor list even if they are not part of the initial target set of the random walk. This guarantees that important connections are reflected in the graph structure, enhancing overall connectivity.

## Dynamic temporal analysis module

In the context of accelerated information dissemination and short public attention spans, controlling the credibility and dissemination paths of news has become increasingly challenging, especially during emergencies. This section analyzes news dissemination patterns through temporal dynamic features and propagation information to enhance transparency and controllability.

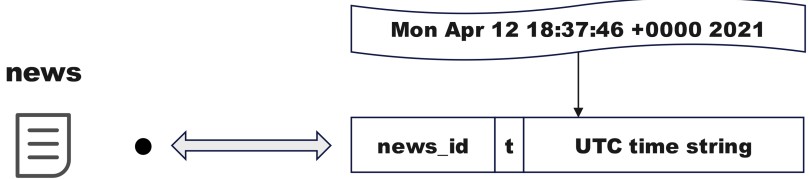

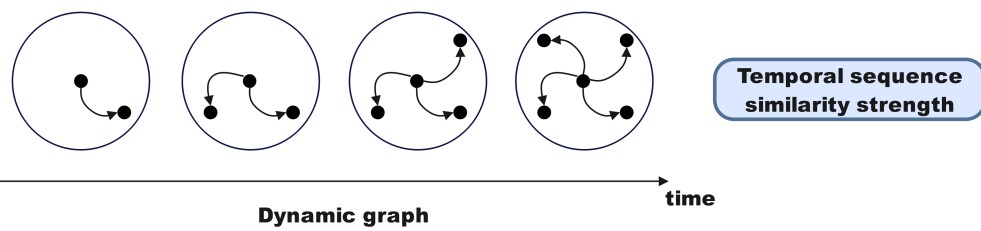

**Figure 5** Temporal sequence penetration fusion module.

*Temporal sequence penetration fusion module*

As the internet and social media continue to evolve, focusing on the time dimension is crucial for studying news dissemination. By analyzing the temporal proximity of nodes, the concept of time similarity intensity is introduced to measure the impact of related news and posts on information spread. The core idea is that the closer the time, the stronger the correlation in information dissemination, enabling a more accurate assessment of the cumulative effects and diffusion paths of information sources.

i. **Incorporating time information into attribute features and graph structure**

As shown in Fig. 5, convert the time string (*e.g.*, Mon Apr12 18 : 37 : 46 2021) into a UTC timestamp to embed temporal information into the identifiers of graph nodes and edges, integrating time data into the graph structure. Node identifiers embed time by concatenating the news ID with the timestamp (*e.g.*, $news_{id} + '' t'' +$ changetime(source$_{tweet}$['created_at'])). Similarly, edge identifiers incorporate reply hierarchy relationships and timestamps to associate news with posts or posts with other posts in the graph structure. Even if replies share the same news ID, they can be distinguished by their timestamps. In the final graph file, all relationships between nodes and edges include temporal information, ensuring that the graph structure's relationships are closely linked to time.

ii. **Enhancing the target news node with temporal similarity based on neighbor features**

By leveraging neighbor features and time similarity, the target news node is enhanced to capture the dynamics of news dissemination in complex networks. Temporal similarity is determined by measuring the interval separating the $x - th$ neighbor and the target news $n_i$ in its neighborhood $N^{n_i}$ and the target news $n_i$ itself.

The time interval $I_x^n$ is calculated according to the following formula:

$$I_x^n = \frac{\exp(t_x - t_{n_i})}{\sum_{j=1}^{k_n} \exp(t_j - t_{n_i})}.$$

Here, $t_{n_i}$ the publishing timestamp of the target news, $t_x$ indicates the publication moment of the $x - th$ neighbor within the neighborhood $N^{n_i}$ associated with $n_i$, while $k_n$ refers to the count of news neighbors associated with $n_i$. Nodes that are temporally closer exhibit stronger correlations in information dissemination and greater content relevance. Therefore, we assign higher weights to such nodes. The temporal similarity strength $S_n^x$ between a neighboring node in $N^n$ and the target news $n_i$ is defined by the following formula:

$$S_x^n = \frac{\exp(-I_x^n)}{\sum_{j=1}^{k_n} \exp(-I_j^n)}.$$

Here, $I_x^n$ represents the time gap between the publication of the $x - th$ neighbor in $N^n$ and the target node $n_i$. $k_n$ denotes the total count of neighbors associated with $n_i$.

Similarly, the temporal similarity strength $S^p$ between posts can be calculated using the same method described above, where $S^n \in \mathbb{R}^{k_n}$ and $S^p \in \mathbb{R}^{k_p}$.

iii. **Concatenating the attribute characteristics of news nodes**

A self-attention mechanism uncovers feature dependencies, enhancing contextual semantics, First, the attribute features of the news node, including its title and content, are processed and combined. Then, they are linearly transformed to a unified dimension $d$, facilitating subsequent neural network computations. The title feature of the neighboring news $E^t$ is given by the following formula:

$$E^t = concat(e_1^t, e_2^t, \ldots\ldots, e_{k_n}^t)W^t.$$

Here, $W^t \in \mathbb{R}^{d_t \times d}$ represents the trainable parameters, where $d_t$ and $d$ denote the unified projection dimension of the title feature and the embedding dimension, respectively. $k_n$ the count of neighboring nodes associated with the target news $n_i$. $concat(\cdot)$ denotes the vector concatenation operation, and $E^t \in \mathbb{R}^{k_n \times d}$.

Each embedding vector undergoes linear transformations through a self-attention layer to generate query, key, and value vectors. Three independent linear layers are initialized to perform matrix operations on the embedding vectors, producing tensors with a shape of $[batch\_size, seq\_len, n\_heads*head\_dim]$ represents the vector dimension for each attention head. The output of the multi-head attention is further processed through a linear layer to produce tensors of the same shape. This operation integrates the attribute features from the news titles of neighboring nodes and employs a multi-head attention mechanism to refine the current news node's attributes, enhancing its semantic dependencies and structural features. The semantic relational features of the news title $E_{rel}^t$ are calculated using the following formula:

$$E_{rel}^t = MultiHead(Q, K, V)$$
$$= MultiHead(E^t, E^t, E^t).$$

Here, $E_{rel}^t \in \mathbb{R}^{k_n \times d}$, the formula for $E_{rel}^t$ means that the input $E^t$ is mapped into three matrices: query, key, and value, through the multi-head attention mechanism. Specifically, it is linearly transformed as $Q' = E^t W_t^Q$, $K' = E^t W_t^K$, and $V' = E^t W_t^V$, where $W_t^Q$, $W_t^K$, and $W_t^V$ represent the weight matrices corresponding to the query, key, and value. The multi-head attention mechanism is calculated using the following equation:

$$MultiHead(Q, K, V) = ( \overset{h}{\underset{i=1}{\|}} Attention(QW^Q, KW^K, VW^V))W^M$$

$$Attention(Q', K', V') = softmax(\frac{Q'K'^T}{\sqrt{d_k}})V'.$$

Here, $d_k$ represents the dimension of $K'$. $W^Q \in \mathbb{R}^{d \times \frac{d}{h}}$, $W^K \in \mathbb{R}^{d \times \frac{d}{h}}$, and $W^V \in \mathbb{R}^{d \times \frac{d}{h}}$. $d$ indicates the unified projection size, and $\|$ signifies the operation of joining features. Using a comparable method described above, the semantic relational features of the news content $E_{rel}^c$, the visual semantic relational features $E_{rel}^v$, and the temporal semantic relational features $E_{rel}^{s^n}$ for neighboring nodes can also be obtained.

Time information from neighboring nodes is integrated into the attribute features to capture their dynamic relationships. Subsequently, varying weights are assigned to each neighbor. In real social networks, the process of news dissemination gives varying levels of attention to different neighboring nodes, enabling the dynamic capture of the social structure information of the news.

iv. **Temporal weighted diffusion features**

Finally, the fused feature representation for each node is obtained. The calculation formula for the temporal diffusion features of the news title is as follows:

$$E_{tdf}^t = MultiHead(Q, K, V) = MultiHead(E_{rel}^{s^n}, E_{rel}^t, E_{rel}^t).$$

Here, $E_{rel}^{s^n}$ and $E_{rel}^t$ denote the temporal semantic relationships and the title-related semantic relationships associated with the main news. $E_{tdf}^n \in \mathbb{R}^{k_n \times d}$ denotes the temporal diffusion attribute of the focus news. $k_n$ denotes the count of neighboring nodes linked to the target news $n_i$. Using the above formula, we can obtain the temporal diffusion features of the news content $E_{tdf}^c$.

Based on the node types (news, user, post), we process the embeddings of different types of neighboring nodes, apply the attention mechanism, and fuse temporal and content information, followed by processing with a bidirectional recurrent neural network (Bi-RNN). The embeddings are then aggregated using mean pooling, and the pooled result is returned. Finally, the fused features of the news node are obtained, and the calculation formula is as follows:

$$E_{fus}^n = Fusion_{Bi-RNN}\left(E_{tdf}^t, E_{tdf}^c, E_{tdf}^c\right) = MeanPool(E_{tdf}^t, E_{tdf}^c, E_{tdf}^c).$$

Here, $E_{fus}^n \in \mathbb{R}^{k_n \times d}$, where $k_n$ refers to the count of connected nodes related to the main news $n_i$. $MeanPool(\cdot)$ denotes the operation of mean pooling.

Integrating the graph structure enhancement module, we extract fused features for users, news, and posts in the heterogeneous graph. Using the temporal sequence permeation

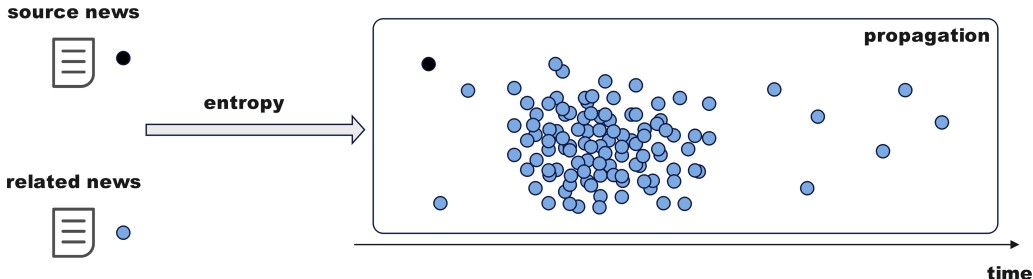

**Figure 6** Temporal sequence penetration fusion module.

fusion module, we dynamically capture the contextual semantics of neighboring nodes within the social network, integrating time-related information. When calculating the fused features of post nodes, temporal permeation is not applied to user neighbors as there are no dynamic relationships between users. By combining semantically enhanced user features with other user attributes, the fused features of post nodes $E_{fus}^{p}$ are obtained. Finally, the fused features for target news nodes $E_{fus}^{n} \in \mathbb{R}^{1 \times d}$, post nodes $E_{fus}^{p} \in \mathbb{R}^{k_p \times d}$, and user nodes $E_{fus}^{n} \in \mathbb{R}^{k_u \times d}$ are computed, where $k_p$ and $k_u$ indicate the quantities of post-type and user-type neighbors associated with $n_i$, and $d$ refers to the dimensionality of the projection.

### Dynamic monitoring in the temporal dimension

The above module optimizes fake news detection through complex networks and information dissemination. This subsection delves deeper from sociological and psychological perspectives. As shown in Fig. 6, fake news often draws inspiration from mainstream public opinion, using high-exposure content to attract attention, with more concentrated release and dissemination times. Monitoring the propagation disorder of rumors and non-rumors using entropy enhances the model's detection accuracy and timeliness.

i. **Calculation of dissemination disorder**

Extract the source tweet's publication time for each news item from the dataset as the starting point and record the reaction tweets' publication times to form a time series. Define time intervals (*e.g.*, 1 h, 2 h, 6 h, 12 h, or even up to 24 h, *etc.*) starting from the source tweet time, segment the timeline, and count the number of tweets in each interval to create a propagation distribution and calculate entropy. A smaller entropy indicates concentrated propagation, while a larger entropy suggests more dispersed or disordered propagation.

$$H = -\sum_{i=1}^{k_p} p(x_{p_i}) \log p(x_{p_i}).$$

Here, $p(x_{p_i})$ represents the probability distribution of a specific post within each time period. $k_p$ represents the count of post-type neighbors linked to $n_i$. By comparing the entropy changes of rumors and non-rumors, the differences in their propagation patterns

can be analyzed. Low entropy indicates that rumors tend to spread rapidly in a short period, while high entropy reflects the sustained diffusion of real news across multiple nodes. In the early stages of propagation, the low number of tweets may cause entropy calculations to be affected by sparsity. This can be addressed using weighted entropy and smoothing techniques.

ii. **Introducing disorder as a temporal feature into the model**

Propagation entropy, as a measure of news dissemination disorder, can be combined with other features and input into the model to enhance news authenticity detection. Additionally, calculating entropy at different time intervals (*e.g.*, every 6 or 12 h) forms a time series that can be input into a Transformer temporal model to capture propagation dynamics. This approach better reflects the timeliness and complexity of propagation paths, thereby optimizing the effectiveness of fake news detection.

## Global semantic analysis module

Building on the graph structure enhancement module and dynamic temporal analysis module described above, this section introduces a module that optimizes the graph structure through dynamic attention coefficients. This module uses a graph attention mechanism, combining temporal semantics and the dynamic relationships of neighboring nodes, to calculate positive and negative attention coefficients and integrate node features. Meanwhile, the model employs a multi-modal input and feature fusion strategy, leveraging the attention mechanism, the model effectively combines the attributes of news nodes and their neighboring nodes, enhancing the precision of fake news detection. The design of the graph alignment module is depicted in Fig. 1.

### *Adding dynamic attention coefficients to optimize the graph structure*

Temporal semantics are added to dynamic neighbor embeddings, and attention coefficients are computed considering node relationships. This means that the interaction between nodes depends not only on their static topology but also on their temporal interactions. The dynamic attention coefficient integrates temporal semantics, enabling the model to more accurately capture node dependencies and information flow at different time points. The formula for the dynamic attention coefficient $\delta_+$ is as follows:

$$\delta_+ = LeakyReLU((E_{fus}W_{\alpha_1}) + (E_{fus}W_{\alpha_2})^T).$$

Here, $E_{fus}$ represents the fused features of the news node, and $W$, $\alpha_1$ and $\alpha_2$ are trainable parameters. The adjacency matrix $A$ is divided into upper $d$ rows and lower $d$ rows, corresponding to $\alpha_1$ and $\alpha_2$. $\alpha \epsilon \mathbb{R}^{2d}$, $E_{fus} \in \mathbb{R}^{l \times d}$, $W \in \mathbb{R}^{d \times d}$, and $\delta_+ \in \mathbb{R}^{l \times l}$.

Subsequently, the dynamic attention coefficients are standardized using the following formula:

$$\delta'_+ = softmax(\delta_+).$$

We use the softmax function for normalization because some posts express opposition to the news being released, which means that some values of $\delta_+$ might be negative. However, after applying softmax brings values closer to zero, reducing the influence of negative ones.

Let $\delta_- = -\delta_+$, and the normalization formula is as follows:

$$\delta_-' = softmax(\delta_-).$$

The graph attention mechanism performs a weighted calculation on the nodes in the embedding sequence by comprehensively analyzing the temporal context of each node and the dynamic relationships with its neighboring nodes. This dynamic weighting calculation not only reflects the immediate associations between nodes but also considers their interactions at different time steps, thereby improving the timeliness and accuracy of feature representation. By concatenating the aggregated features of both positive and negative correlations, the output features processed by the fully connected layer are represented as follows:

$$f_g = ReLU((\delta_+' E_{fus} || \delta_-' E_{fus}) W_g).$$

Here, $E_{fus}$ represents the fused features of the news node, and $W_g$ is a trainable parameter, with $W_g \in \mathbb{R}^{2d \times d}$. $ReLU(\cdot)$ is used as a nonlinear function, $||$ denotes a joining operation, and $f_g \in \mathbb{R}^{l \times d}$.

During model training, different dynamic attention coefficients are produced. Based on the positive attention weights, the information from neighboring nodes is weighted and summed to generate updated representations for individual nodes. Similarly, the feature representation for negative attention is also derived. The node features under both positive and negative attention are concatenated to form a feature matrix that is twice the original size. A new weight matrix, used in a variant of the graph attention network (GAT), is then applied to the concatenated features for m linear transformations, generating new node features. This process yields a more comprehensive and integrated graph structure feature $F_g$. The calculation formula is as follows:

$$F_g = \sigma_{elu}(GAT(\overset{m}{\underset{i=1}{||}} f_g^i)).$$

Here, $f_g^i$ represents the graph structural features obtained in the $i-th$ iteration, where $i \in [1, h]$. $GAT(\cdot)$ represents the graph attention module, $\sigma_{elu}(\cdot)$ stands for the *elu* activation function, $||$ is the concatenation operation, and $F_g \in \mathbb{R}^{l \times d}$.

### Multimodal feature integration and prediction

This module aggregates the features of news nodes and their neighbors (users, posts) using a Transformer, integrating temporal features, node entropy weights, and fused visual-textual representations to enhance the perception of temporal dynamics and the complexity of multimodal node information. Utilizing the multi-head attention mechanism, it captures complex interactions between nodes and across modalities, generating enriched node representations for prediction. The classification process uses cross-entropy loss for training and leverages Adam for optimization.

   i. **Multimodal input and fusion strategy:** The input includes multimodal data such as text, visual content, user information, and the level of news propagation disorder. Features from text and image modalities are fused through a visual-textual embedding

**Table 1** Statistical data of the dataset.

| Statistics | GossipCop | PHEME |
|---|---|---|
| Total news | 20,359 (R:15446/F:4913) | 6,425 (R:4023/F: 2402) |
| Users | 429,628 | 51,043 |
| Posts | 1,192,766 | 98,929 |
| Total nodes | 1,642,753 | 156,397 |

mechanism, while user and propagation features are concatenated at the input stage. All modalities are then unified and weighted in the input layer to form a comprehensive feature representation for model training. The news propagation disorder feature is treated as an independent channel that captures abnormal dissemination patterns, and is integrated with other modality outputs at the decision layer to generate the final classification result.

2. **Feature encoding and attention mechanism:** The multimodal node features are further processed using a Transformer, incorporating node type and positional encoding. The multi-head attention mechanism dynamically assigns weights to different modality channels—including textual, visual, user profile, and propagation disorder features—automatically learning their relative importance for fake news detection. This facilitates the extraction of deep semantic dependencies across both content and structure, thereby enhancing model accuracy.

3. **Prediction and classification:** The final node embeddings are passed through a linear layer and an activation function, producing classification outputs for identifying fake news. The activation function is defined as:

$$\sigma(z) = \frac{1}{1 + e^{-z}}.$$

The loss function uses binary cross-entropy combined with L2-normalized vectors to optimize the model and generate the final prediction probabilities for rumors or non-rumors.

## EXPERIMENT

### Dataset

To study the impact of news propagation patterns, user interactions, and temporal dynamic features on fake news detection, we selected the GossipCop and PHEME datasets. These datasets provide rich multimodal information, demonstrating how fake news spreads across various contexts while providing strong support for our research. An overview of the dataset statistics is shown in Table 1.

- **GossipCop dataset:** Originating from FakeNewsNet, this dataset analyzes the dissemination of authentic and misleading news in the entertainment field. It incorporates diverse modalities, including textual content, visual data, user interactions, and dissemination paths. The propagation path information allows us to utilize the feature representation module and graph structure enhancement module to extract

static and dynamic attributes of nodes, analyze complex propagation dependencies, and reveal the distinctive propagation patterns of fake news in the entertainment domain.

- **PHEME dataset:** Focused on rumor dissemination during emergency events, particularly on the Twitter platform, this dataset includes information such as text, timestamps, user interactions, and propagation paths. It supports temporal feature analysis by measuring dynamic temporal characteristics and, through the temporal dynamics fusion module, captures the temporal dynamics and anomalous patterns of fake news dissemination using temporal similarity and self-attention mechanisms.

## Experimental setup

The two datasets are divided into 70% for training, 10% for validation, and 20% for testing. In the text feature representation module, the RoBERTa model handles the text data from the datasets, followed by the T5 model, which generates embedding vectors with a 768-dimensional output. In the graph structure enhancement module, for heterogeneous graph optimization using the Random Walk with Restart (TWR) method, the PHEME dataset uses a maximum number of steps ($\max_{steps}$) of 10,000, a maximum number of neighbors ($\max_{neigh}$) of 200, and a restart rate ($restart_{rate}$) of 0.5 for each step returning to the starting node. In the GossipCop dataset, the maximum number of steps ($\max_{steps}$) is 10,000, the maximum number of neighbors (($\max_{neigh}$) is 50, and the restart rate ($restart_{rate}$) is set to 0.5 for each step.

For the temporal sequence fusion module and the graph modality alignment module, the multi-head attention mechanism uses $h = 8$ attention heads. In the temporal sequence fusion module, a dropout rate of 0.2 is applied in the attention mechanism for both the PolitiFact and GossipCop datasets. In the temporal monitoring module, the entropy time step is set to 12. For integrating graph structure features, the position embedding encoding uses a dropout rate of 0.1, and the Transformer model has one encoder–decoder layer for PHEME and GossipCop. The optimal results are chosen from five independent trials. Each model undergoes training for a maximum of 40 epochs, using a patience parameter set to 5.

## Evaluation metrics

To evaluate the performance of the methods, we use precision, recall, and F1-score. Accuracy, described as the ratio of correct predictions to total samples, is also considered. Although it is a common metric for fake news detection, it may not fully reflect performance due to potential data imbalances. As a result, precision, recall, and F1-score are used to separately assess the predictions of real and fake news. Here, P and N denote real and fake news instances, while T and F indicate the model's predictions for each. These metrics provide a more balanced evaluation of classification on imbalanced datasets.

- **Precision**

$$Precision = \frac{TP}{TP + FP}$$

- **Recall**

$$Recall = \frac{TP}{TP + FN}$$

- **F1**

$$F1 = \frac{2 \times Precision \times Recall}{Precision + Recall}.$$

## Baselines

To showcase the capabilities of the DTN model in identifying fake news, we selected multiple reference models for comparison:

**Similarity-Aware Multimodal Prompt Learning (SAMPLE)** (*Jiang et al., 2023*): Introduces similarity-aware multimodal prompt learning, combining prompt templates and adaptive fusion to mitigate cross-modal noise and enhance detection across diverse settings.

**EmotionAware Multimodal Fusion Prompt LEarning (AMPLE)** (*Xu et al., 2024*): Incorporates emotion-aware analysis and hybrid prompts to fuse textual sentiment with multimodal data, improving fake news detection in both few-shot and full-data scenarios.

**COOLANT** (*Wang et al., 2023*): Leverages cross-modal contrastive learning and guided attention to enhance fine-grained image-text alignment, achieving strong performance on benchmark datasets.

**Multi-reading habits fusion reasoning networks (MRHFR)** (*Wu, Liu & Zhang, 2023*): Mimics human reading habits to guide multimodal fusion and inconsistency reasoning, capturing deep semantic correlations and cross-modal contradictions.

**Human Cognition-based Consistency Inference Networks (HCCIN)** (*Wu et al., 2024*): Models human cognition by aligning image-text content, discovering comment clues, and reasoning about consistency for robust multimodal fake news detection.

**Multi-modal Feature-enhanced Attention Networks (MFAN)** (*Zheng et al., 2022*): Employs GANs to integrate text, images, and social graphs, achieving high accuracy through deep multimodal feature interaction.

**Heterogeneous Transformer (HetTransformer)** (*Li et al., 2022*): Applies Transformer architecture to heterogeneous graphs, modeling global semantics and propagation patterns for misinformation detection.

**Text-Clustering Graph Neural Network (TCGNN)** (*Li & Li, 2024*): Constructs graphs purely from textual clustering, capturing fine-grained semantic relations without relying on user or propagation data.

**Multimodal interaction and graph contrastive learning network (MIGCL)** (*Cui & Shang, 2025*): Combines cross-modal alignment with graph contrastive learning to model intra- and intermodal dynamics, enhancing robustness in multimodal fake news detection.

## Hyperparameter experiments

During the model training process, the learning rate and batch size are two crucial hyperparameters that significantly affect the convergence speed, stability, and final performance of the model. Proper selection of hyperparameters can speed up training and enhance both prediction accuracy and generalization. To investigate how these hyperparameters affect the DTN model, we performed experiments on learning rate and batch size, analyzing their influence on the PHEME and GossipCop datasets. These

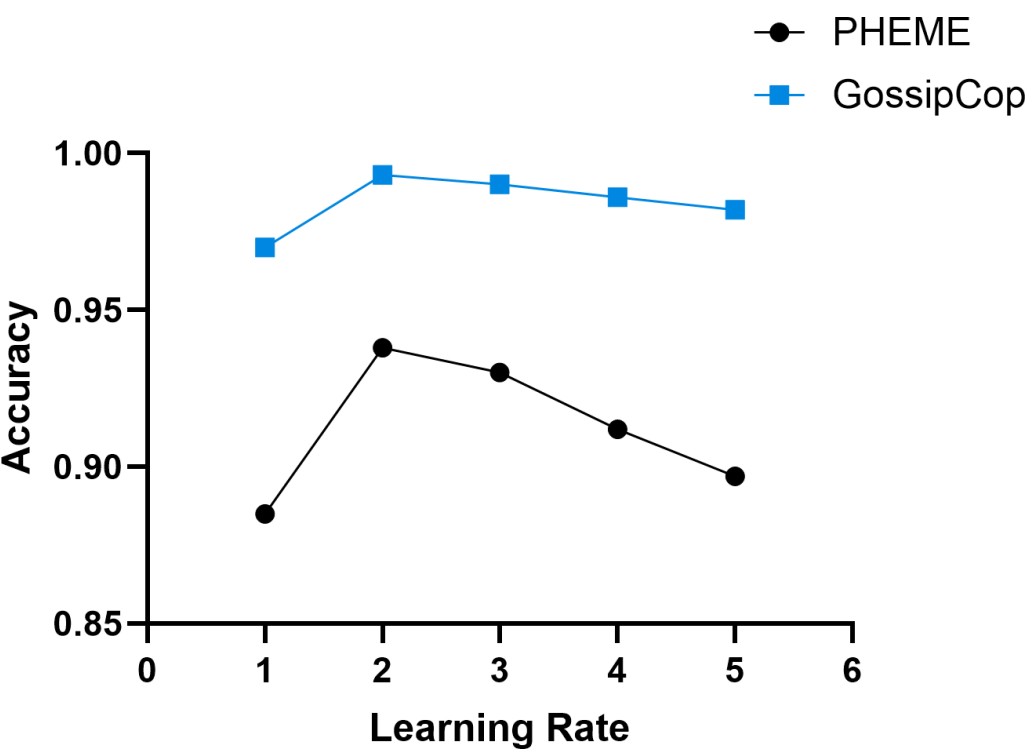

**Figure 7** The effect of varying learning rates on the model's training performance.

experiments aim to refine the model's performance and improve its generalization capabilities.

### Learning rate

The learning rate is a key hyperparameter that significantly impacts model training performance, including convergence speed, stability, and overall effectiveness. A larger learning rate can accelerate early parameter updates but may cause oscillations or miss the optimal solution, leading to non-convergence or degraded performance. A smaller learning rate ensures stable updates but slows down training and may get stuck in local optima, limiting performance improvement. In the experiments, the value of the learning rate $\alpha$ was assigned a value of $10^{-i}$, where $i \in [1, 5]$. Figure 7 presents the experimental results.

Overall, the results demonstrate that batch size significantly impacts DTN performance, and selecting an appropriate size based on dataset complexity and sample volume is essential to balance convergence speed and detection accuracy.

### Batch size

In deep learning, batch size serves as an essential hyperparameter that impacts training efficiency. Varying batch sizes impact both gradient estimation accuracy and memory consumption. A smaller batch size enables more frequent parameter updates and introduces stochastic gradient estimates, which can help the model avoid local optima. However, excessively small batch sizes may result in instability, slowing down convergence and

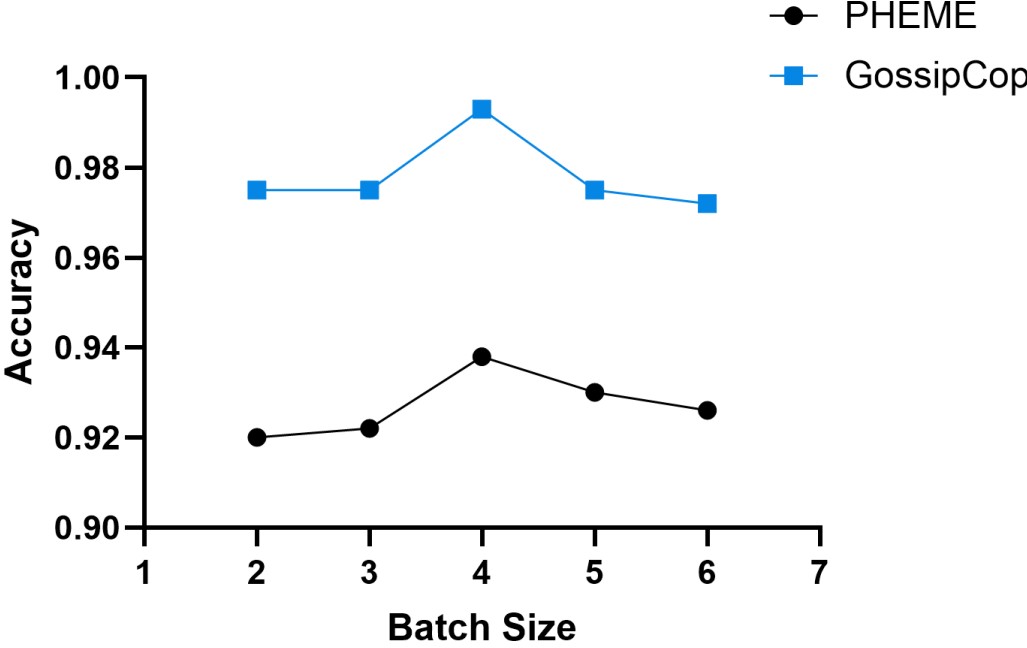

**Figure 8** **The effect of varying batch size on the model's training performance.**

degrading final performance. Conversely, using larger batch sizes improves gradient estimation accuracy and training stability, but it reduces update frequency, resulting in longer durations and higher memory consumption.

In our experiments, the batch size was set as $2^i$, where $i \in [2, 6]$. The results, illustrated in Fig. 8, indicate that increasing the batch size improves DTN's training speed significantly. However, when the batch size becomes too large, performance drops, likely due to reduced update frequency, causing the model to miss optimal convergence points. For the PHEME dataset, the highest accuracy of 0.938 is obtained with a batch size of 16, while the GossipCop dataset achieves its peak accuracy of 0.993 at the same batch size.

Overall, the results demonstrate that batch size significantly impacts DTN performance, and selecting an appropriate size based on dataset complexity and sample volume is essential to balance convergence speed and detection accuracy.

## Results and Discussion

We will address the three core questions proposed in the introduction through our experimental results:

**Q1:** How can we effectively capture the temporal dynamics of nodes?

**Q2:** How can we dynamically fuse multi-modal information to fully leverage the complementarity between modalities?

**Q3:** How can we reveal the anomalous patterns of fake news in temporal and spatial distributions to enhance detection accuracy?

**Table 2  Performance of different methods on PHEME and GossipCop datasets.**

| Dataset | Method | Accuracy | True news | | | Fake news | | |
|---|---|---|---|---|---|---|---|---|
| | | | Precision | Recall | F1 | Precision | Recall | F1 |
| PHEME | SAMPLE | 0.803 | 0.820 | 0.816 | 0.810 | 0.797 | 0.765 | 0.812 |
| | AMPLE | 0.852 | 0.857 | 0.869 | 0.843 | 0.812 | 0.798 | 0.852 |
| | COOLANT | 0.868 | 0.862 | 0.856 | 0.859 | 0.804 | 0.818 | 0.811 |
| | MRHFR | 0.811 | 0.818 | 0.805 | 0.811 | 0.814 | 0.792 | 0.821 |
| | HCCIN | 0.904 | 0.916 | 0.930 | 0.919 | 0.846 | 0.861 | 0.853 |
| | TCGNN | 0.867 | 0.841 | 0.826 | 0.833 | 0.794 | 0.809 | 0.801 |
| | MFAN | 0.893 | 0.997 | 0.863 | 0.925 | 0.689 | **0.992** | 0.814 |
| | HetTransformer | 0.825 | 0.868 | 0.849 | 0.858 | 0.756 | 0.784 | 0.770 |
| | MIGCL | 0.898 | 0.881 | 0.855 | 0.868 | **0.908** | 0.895 | 0.917 |
| | **DTN** | **0.938** | **0.967** | **0.964** | **0.936** | 0.895 | 0.913 | **0.924** |
| GossipCop | SAMPLE | 0.640 | 0.650 | 0.600 | 0.620 | 0.630 | 0.640 | 0.620 |
| | AMPLE | 0.850 | 0.820 | 0.780 | 0.800 | 0.780 | 0.820 | 0.800 |
| | COOLANT | 0.915 | 0.895 | 0.893 | 0.894 | 0.885 | 0.886 | 0.885 |
| | MRHFR | 0.928 | 0.930 | 0.926 | 0.928 | 0.918 | 0.920 | 0.919 |
| | HCCIN | 0.926 | 0.920 | 0.927 | 0.923 | 0.892 | 0.910 | 0.901 |
| | TCGNN | 0.922 | 0.911 | 0.924 | 0.917 | 0.902 | 0.906 | 0.904 |
| | MFAN | 0.778 | 0.825 | 0.892 | 0.858 | 0.578 | 0.439 | 0.499 |
| | HetTransformer | 0.990 | **0.994** | **0.993** | **0.993** | 0.978 | 0.980 | 0.979 |
| | MIGCL | 0.945 | 0.928 | 0.944 | 0.838 | 0.924 | 0.926 | 0.925 |
| | **DTN** | **0.993** | 0.992 | 0.989 | 0.989 | **0.982** | **0.982** | **0.989** |

**Notes.**
The best results for each metric are highlighted in bold.

### Overall model performance

Table 2 presents the comparative performance of a series of representative models on the PHEME and GossipCop datasets, with evaluation metrics covering accuracy, precision, recall, and F1-score for both real and fake news classification tasks. The best results for each metric are highlighted in bold. The visualized comparison results are shown in Figs. 9 and 10. The selected baselines encompass a diverse spectrum of methodological paradigms that collectively reflect the current landscape of fake news detection. Specifically, multimodal approaches such as SAMPLE, AMPLE, COOLANT, and MFAN exploit complementary textual and visual information to capture enriched semantic representations; graph-based models including TCGNN, MRHFR, and HCCIN emphasize relational and topological structures within content dissemination or user interaction networks; while hybrid architectures like HetTransformer and MIGCL integrate multimodal cues with graph-based reasoning to simultaneously leverage content semantics and structural dependencies. The inclusion of these diverse and competitive baselines not only facilitates a rigorous and multidimensional evaluation but also highlights the importance of jointly modeling

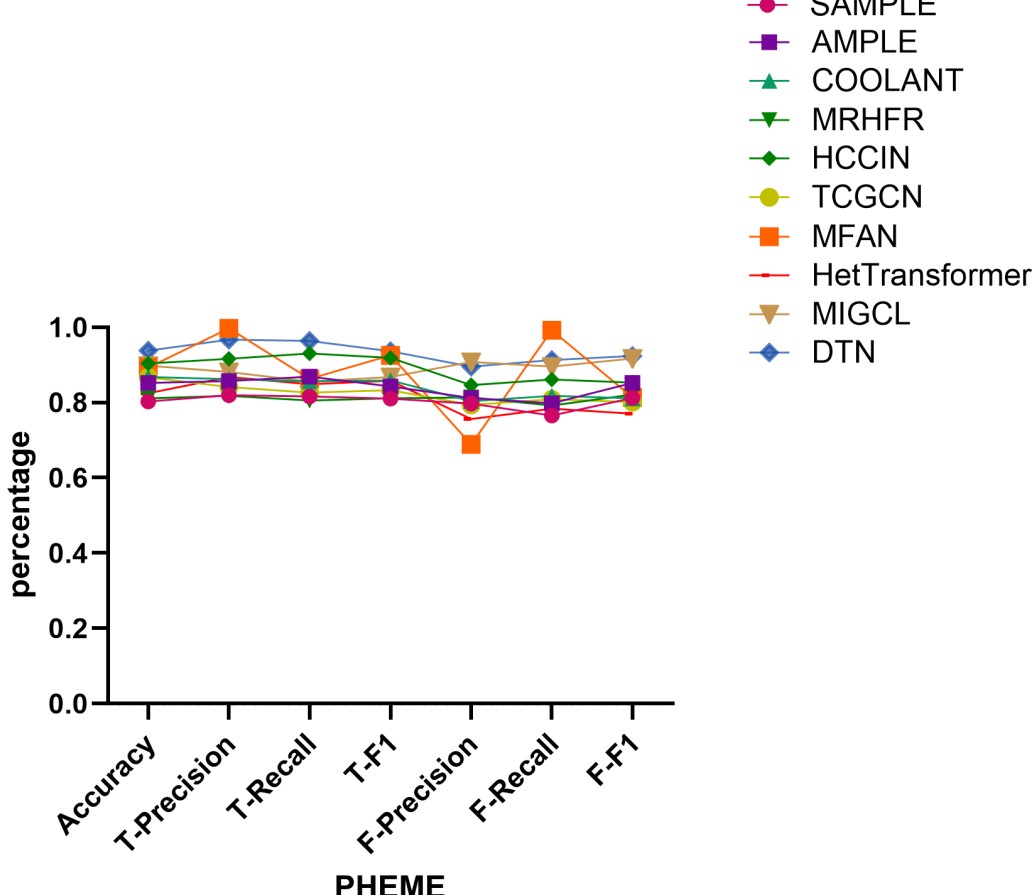

**Figure 9** Comparison of model performance on the PHEME dataset.

multimodal semantics and structural context, which provides a comprehensive foundation for validating the effectiveness and generalizability of our proposed framework.

**Performance on the PHEME dataset.** On the PHEME dataset, the DTN model achieves the highest overall performance, with an accuracy of 0.938. For real news classification, it obtains precision, recall, and F1-score values of 0.967, 0.964, and 0.936, respectively, outperforming competitive baselines such as MFAN (0.997, 0.863, 0.925) and HCCIN (0.916, 0.930, 0.919).

Regarding fake news classification, DTN achieves precision, recall, and F1-score values of 0.895, 0.913, and 0.924, respectively, surpassing all other methods. Although MIGCL yields comparable results (0.908, 0.925, 0.917), DTN demonstrates a more balanced performance across metrics, reflecting its robustness in distinguishing both real and fake news.

**Performance on the GossipCop dataset.** On the GossipCop dataset, DTN achieves the highest accuracy of 0.993. For real news, it records precision, recall, and F1-score values of 0.992, 0.989, and 0.989, respectively. While HetTransformer attains a marginally higher precision of 0.994, its recall and F1-score (both at 0.993) remain comparable, and DTN exhibits greater consistency across both classes. For fake news classification, DTN again

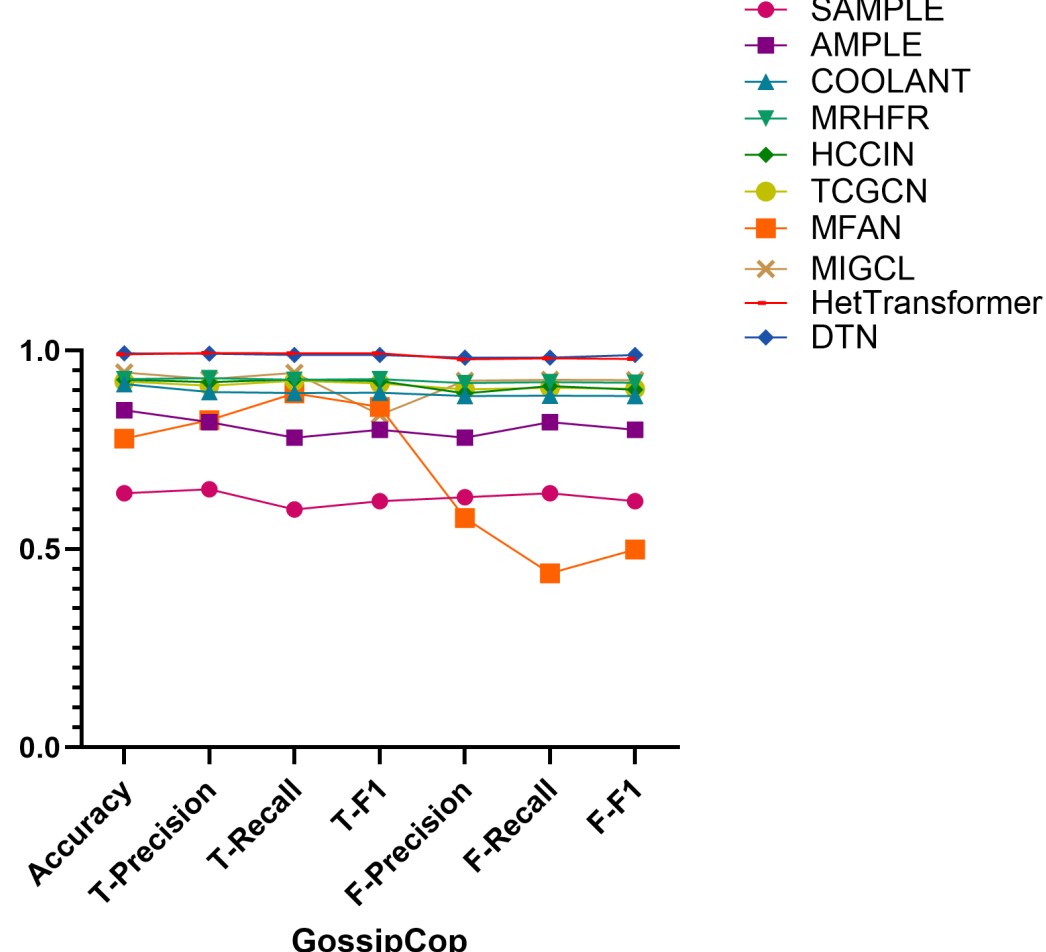

**Figure 10** Comparison of model performance on the GossipCop dataset.

leads with precision, recall, and F1-score values of 0.982, 0.982, and 0.989, respectively. In comparison, MIGCL achieves scores of 0.964, 0.985, and 0.974, indicating that DTN maintains superior balance and overall effectiveness.

**Summary.** The results on both datasets confirm the superior performance of DTN compared to unimodal, multimodal, and hybrid baselines. While existing models such as MFAN, MIGCL, and HCCIN demonstrate strength in specific aspects, DTN consistently achieves competitive scores across all metrics. Its performance benefits from a unified framework that incorporates temporal dynamics, multimodal fusion, and structural reasoning, resulting in a more accurate, stable, and generalizable solution for fake news detection.

## Question 1: Effectively capturing the temporal dynamics of nodes

In the real world, network topology often exhibits dynamic characteristics, with *Jin et al. (2023)*, *Zheng et al. (2023)* and *Chen et al. (2024)* indicating that the dynamic changes between nodes are comparable to the communication timescale among them. We capture

the temporal dynamics of news nodes by constructing a nested dictionary structure, incorporating the publication time and reply relationships into a hierarchical structure to reflect the temporal order and hierarchical relationships between nodes. In this structure, timestamps are part of each node's identifier, recording the specific time position of each node (post) within the propagation chain. When a new reply post appears, the structure tree dynamically updates, and the new node's position is automatically adjusted according to its time and reply relationship. This dynamic insertion and hierarchical construction method enables comprehensive capture of post node changes and temporal characteristics throughout the propagation process.

The experimental findings highlight that our DTN method outperforms the HetGNN method in capturing temporal dynamics. DTN dynamically adjusts the time window to capture the time differences between nodes, while HetGNN uses a fixed time interval, making it difficult to capture subtle temporal variations between nodes. Consequently, DTN exhibits significant improvements in accuracy, precision for real news, F1-score, as well as recall and F1-score for fake news, further validating the effectiveness of dynamic time windows in enhancing model accuracy.

## Question 2: Dynamic fusion of multi-modal information and utilization of complementarity

Multimodal information fusion incorporates various types of data, including textual content, videos, audio signals, social networks, and temporal information. Our DTN method focuses on the fusion of text, social network, and time modalities (*Zhu et al., 2024*; *Zhang et al., 2025*) to comprehensively capture the propagation characteristics of fake news. In the text modality, DTN extracts not only the surface-level semantic information of news content but also delves into contextual associations and sentiment inclinations, enhancing the deep understanding of news semantics. In the social network modality, DTN integrates interaction relationships and information propagation paths between nodes, constructing global and local semantic information that includes node relationships using dynamic social data. This structure enables the model to grasp news propagation patterns and interaction features within the network comprehensively. Simultaneously, the introduction of the time modality allows the model to track the time sequence and evolution of information dissemination, dynamically analyzing the temporal propagation patterns of nodes.

These multi-modal features, after fusion processing, are input into the Transformer encoder, enabling more precise information representation while preserving dynamic interaction features and the temporal sequence of nodes within the propagation path. Compared to the BERT model, which solely focuses on textual semantics, DTN not only performs semantic analysis but also accounts for the propagation patterns of information within the network as they change over time. Furthermore, in contrast to models like IARNET, HMGNN, HGT, and HetTransformer that focus on learning complex structures and node information within heterogeneous graphs, DTN dynamically fuses multi-modal features, resulting in more stable and superior model performance.

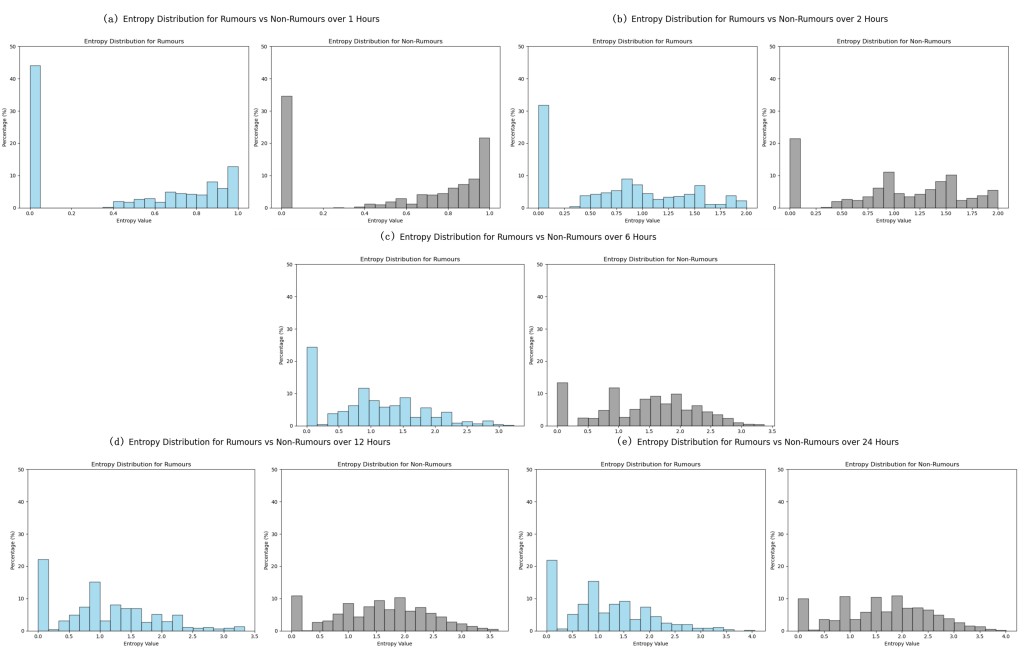

**Figure 11** Dispersion values of news dissemination in different time ranges on the PHEME dataset.

## Question 3: Revealing anomalous spatiotemporal patterns of fake news

In the analysis of news dissemination time series, different time steps $\Delta t$ can reveal various patterns and characteristics of the propagation process. When the time step is small (*e.g.*, 1 h, 2 h), it can capture the rapid spread of news within a short time, where the concentration of dissemination is high, meaning the number of posts is often concentrated within a small period following the news release, reflecting a lower entropy. As the time step increases (*e.g.*, 6 h, 12 h, 24 h), the long-term trend of dissemination becomes more apparent, with posts being distributed over a longer time period and entropy gradually increasing. This indicates that news dissemination becomes more dispersed over longer timescales, potentially exhibiting a long-tail effect where the spread of posts continues well beyond the initial release.

By analyzing and comparing the dissemination disorder under different time ranges, we set the $t_{range}$ to 1 h, 2 h, 6 h, 12 h, and 24 h, with $s_{le}$ set to 30 min.

As shown in Fig. 11, the $X$-axis represents the entropy of news dissemination, where higher entropy indicates greater disorder in dissemination, and the $Y$-axis represents the proportion of each entropy value within the dataset, reflecting the prevalence of specific dissemination characteristics.

**Within 1 h:** As shown in Fig. 11A, rumor entropy is concentrated in the low range (0.0), with a small presence in the high range (0.8–.0), indicating concentrated paths and high certainty in early stages. Non-rumor entropy is also concentrated at 0.0 but has fewer high-entropy values, showing slightly lower concentration than rumors.

**After 2 h:** As shown in Fig. 11B, rumor entropy spreads from 0.0 to 0.5−1.5, with paths and patterns diversifying and uncertainty increasing. Non-rumor entropy remains concentrated around 0.0 and 1.0, with lower uncertainty than rumors.

**Within 6 h:** As shown in Fig. 11C, rumor entropy is primarily distributed between 0.5−1.5, with further diversification of dissemination. Non-rumor entropy remains relatively concentrated, despite some increase in uncertainty.

**After 12 h:** As shown in Fig. 11D, rumor entropy expands to a range of 0.0−3.0, peaking at 1.0−2.0, indicating significant diversification in dissemination paths. Non-rumor entropy ranges from 0.0 to 2.5, with limited increase in uncertainty.

**Within 24 h:** As shown in Fig. 11E, rumor entropy reaches a maximum of 4.0, showing a significant increase in uncertainty and complexity of dissemination. Non-rumor entropy peaks at 3.5 but remains relatively concentrated, indicating higher stability in dissemination.

These entropy analysis results reveal that fake news detection can be optimized based on temporal and dispersion characteristics of dissemination. Rumors exhibit higher dispersion and disorder within short time frames (1–2 h), making early detection in this window more effective. Over time, rumor dissemination becomes increasingly complex and harder to control. Detection systems can leverage higher entropy changes and dispersion characteristics to identify potential fake news, particularly over longer time frames (6 h and beyond), which is crucial for preventing large-scale dissemination of rumors.

## Case study

To further validate the interpretability and real-world applicability of the proposed DTN model, we analyze two representative case studies that highlight the contrasting temporal and semantic characteristics of fake and real news propagation. These cases demonstrate the model's ability to dynamically fuse multimodal signals—such as textual semantics, temporal burst patterns, user credibility, and engagement statistics—for accurate veracity classification.

**Case A** (Fig. 12) represents a piece of misinformation falsely attributing a quote to a political figure. Although the source tweet originates from a verified user with over 100 k followers, the subsequent propagation pattern reveals suspicious characteristics. The post quickly receives a series of emotionally reactive replies within the first two hours, forming dense clusters with highly similar and sentimentally charged content. Most responses are authored by unverified accounts with low fan bases, and engagement levels spike abnormally within a short window. The DTN model detects this combination of rapid dissemination, semantic redundancy, and low user credibility as an indicator of misinformation, and accurately classifies the case as **Fake**.

**Case B** (Fig. 13), in contrast, involves a verified source tweet discussing a sensitive social issue using neutral language. The propagation unfolds gradually, with responses spread across a longer time frame and containing more diverse viewpoints. Verified and unverified users participate in a balanced manner, and engagement count remains stable without sudden surges. The semantic content of the replies shows thoughtful discussion rather than

|  | source-tweet | Reaction 1 | Reaction 2 | Reaction 3 | Reaction 4 | Reaction 5 | Reaction 6 | ...... | Reaction n |
|---|---|---|---|---|---|---|---|---|---|
| Tweet ID | 544269221564137472 | 544270627817807872 | 544271078944931840 | 544271334218678272 | 544271793054556160 | 544271852399759361 | 498265441543159809 | ...... | 544271900147343361 |
| Content Summary | "Obama: 'I'm not going to be scrambling jets to get a TV producer!'..." | "Haha. His priorities are clear..." | "That's kind of the point. #liblogic..." | "But he'll do it for a deserter..." | "We have a celebrity in the White House..." | "But he will trade 5 terrorists for a traitor..." | "But he'll release 5 terrorists for a traitor..." | ...... | "But he'll release terrorists for a traitor. #FAIL..." |
| Drawing Or Not | Yes | No | No | No | No | No | No | ...... | No |
| Release Time (UTC) | 2014-12-14 07:14:29 | 2014-12-14 07:45:05 | 2014-12-14 08:20:04 | 2014-12-14 08:49:51 | 2014-12-14 10:24:42 | 2014-12-14 13:24:56 | 2014-12-14 19:25:07 | ...... | 2014-12-15 07:26:20 |
| Δ Time ( min ) | 0 | 30.6 | 65.6 | 95.4 | 190.2 | 370.4 | 730.6 | ...... | 1451.8 |
| User Authentication | Yes | No | No | No | No | No | No | ...... | No |
| Fans Level | 100k+ | 1k+ | 500+ | 5k+ | 1k+ | 2K+ | <100 | ...... | <100 |
| Engagement Count | 24630 | 639 | 878 | 968 | 2000 | 1020 | 500 | ...... | 50 |
| Propagation | Moderate abnormal transmission ⚠ | | | | | | | | |
| model prediction | Fake | | | | | | | | |

**Figure 12** Visualization of temporal diffusion anomaly in fake news propagation (Case 1).

|  | source-tweet | Reaction 1 | Reaction 2 | Reaction 3 | Reaction 4 | Reaction 5 | Reaction 6 | ...... | Reaction n |
|---|---|---|---|---|---|---|---|---|---|
| Tweet ID | 498264107460870144 | 498264249110892545 | 498264398310694912 | 498264605131821056 | 498264759079559168 | 498265017863929857 | 498265441543159809 | ...... | 499054873321553920 |
| Content Summary | "The #Ferguson shooting shows that being passive..." | "#RealTalk homie" | "what are the numbers on black cops shooting white teens..." | "zero..because they know that the dominant society..." | "Where's the negro bed wenches talking about..." | "exactly" | "exactly" | ...... | "None of that was done here..." |
| Drawing Or Not | No | No | No | No | No | No | No | ...... | No |
| Release Time (UTC) | 2014-08-10 13:41:36 | 2014-08-10 14:02:09 | 2014-08-10 14:10:45 | 2014-08-10 14:48:34 | 2014-08-10 15:54:01 | 2014-08-10 19:44:49 | 2014-08-11 02:46:01 | ...... | 2014-08-11 13 :46:59 |
| Δ Time ( min ) | 0 | 20.6 | 29.1 | 67.0 | 132.4 | 363.2 | 724.4 | ...... | 1385.4 |
| User Authentication | Yes | Yes | No | No | No | No | Yes | ...... | No |
| Fans Level | 100k+ | 10k+ | <100 | 500+ | <100 | 500+ | 100K+ | ...... | <100 |
| Engagement Count | 197 | 12 | 26 | 40 | 71 | 20 | 3 | ...... | 0 |
| Propagation | Steady diffusion rhythm 🔔 | | | | | | | | |
| model prediction | True | | | | | | | | |

**Figure 13** Visualization of stable diffusion pattern in real news propagation (Case 2).

coordinated amplification. Capturing these steady temporal signals and heterogeneous user interactions, the DTN model confidently classifies this case as **True**.

Both figures visualize the propagation flow, showing the tweet content, release time, user attributes (*e.g.*, verification, fan level), media presence, engagement count, and the model's prediction. These case studies illustrate the strength of DTN in capturing both temporal dynamics and multimodal semantics, enabling robust detection even under emotionally polarized or information-overloaded settings. By modeling time-sensitive propagation graphs and dynamically fusing multiple signals, DTN enhances detection reliability while maintaining interpretability.

**Table 3  Model performance improvement table.**

| Model | PHEME | GossipCop |
|---|---|---|
| Base model+GSE | 7.20% | 8.10% |
| Base model+TSP | 6.30% | 7.50% |
| Base model+DMT | 5.90% | 6.70% |
| Base model+GSE+TSP | 9.80% | 10.60% |
| Base model+GSE+DMT | 8.90% | 9.70% |
| Base model+GSE+TSP+DMT | 13.20% | 14.50% |

## Ablation study

To improve the effectiveness of the DTN model, we progressively incorporated the graph structure enhancement module (GSE), temporal sequence permeation fusion module (TSP), and temporal dynamic monitoring module (DMT). These two temporal modules are designed from complementary perspectives to enhance the model's sensitivity to time-dependent propagation signals. The specific results are shown in Table 3. Experiments show that each module significantly improves fake news detection capabilities, as detailed below:

## Graph structure enhancement module

Accuracy increased by 7.2% and 8.1% on the two datasets, respectively. This module extracts deeper relationships within propagation networks, particularly capturing complex interaction patterns, thereby improving detection accuracy.

## Temporal sequence permeation fusion module

Accuracy improved by 6.3% and 7.5%, respectively. This module captures temporal dynamics in propagation, enhancing the identification of time-sensitive rumors.

## Temporal dynamic monitoring module

Accuracy increased by 5.9% and 6.7%, respectively. This module captures dynamic changes during propagation, improving the detection of complex propagation patterns.

## Combined module effects

**GSE + TSP:** Accuracy increased by 9.8% and 10.6%, indicating their combination significantly enhances the model's overall capability.

**GSE + DMT:** Accuracy improved by 8.9% and 9.7%, demonstrating their synergy in capturing both static and dynamic features.

**GSE+TSP+DMT (Full DTN):** Accuracy reached 13.2% and 14.5%, the best performance, effectively integrating multidimensional features of news propagation and significantly boosting detection capability.

In summary, progressively incorporating and combining these modules enables the DTN model to excel in detecting fake news within complex propagation networks and time-sensitive contexts, validating the design and contributions of these modules.

## CONCLUSIONS

This paper proposes the DTN model for multimodal fake news detection. By leveraging temporal similarity, the model dynamically weights neighboring nodes in propagation sequences and integrates multimodal information—including text, images, user profiles, and propagation disorder—at the node level. Through temporal-aware social graph modeling, DTN enhances node representation and captures both local and global context in news dissemination. The model also incorporates entropy-based analysis to detect anomalies in propagation patterns, improving detection accuracy. A Transformer encoder is used to model structural semantics and support multimodal feature fusion. Experiments show that DTN consistently outperforms baseline methods across multiple datasets. While our approach effectively integrates multimodal features, it does not explicitly model interactions between modalities. In future work, we plan to explore cross-modal attention mechanisms and contrastive learning strategies to better capture inter-modality correlations. Additionally, we aim to investigate the model's robustness under noisy or adversarial input conditions, and further develop its capability for early-stage detection by analyzing partial cascades in real-time. We also intend to construct time-sensitive and event-driven datasets to support these extensions.

### Funding

This study received funding and support from the following projects: National Natural Science Foundation of China (62472040), Copyright Research Project of China Copyright Protection Center (BQ2024017), Digital Competence and Foreign Language Reading Research (Project No. 27170124040). Research on University Students' Bilingual Reading Efficiency and Teaching Based on Artificial Intelligence and Big Data (Project No. 22150124045). Research Project on Bilingual Reading and Big Data Under Natural Language Contexts (Project No. 20190124076). The funders had no role in study design, data collection and analysis, decision to publish, or preparation of the manuscript.

### Grant Disclosures

The following grant information was disclosed by the authors:
National Natural Science Foundation of China: 62472040.
Copyright Research Project of China Copyright Protection Center: BQ2024017.
Digital Competence and Foreign Language Reading Research: Project No. 27170124040.
Research on University Students' Bilingual Reading Efficiency and Teaching Based on Artificial Intelligence and Big Data: Project No. 22150124045.
Research Project on Bilingual Reading and Big Data Under Natural Language Contexts: Project No. 20190124076.

### Competing Interests

The authors declare there are no competing interests.

## Author Contributions

- Jiaen Hu conceived and designed the experiments, performed the experiments, analyzed the data, performed the computation work, prepared figures and/or tables, authored or reviewed drafts of the article, and approved the final draft.
- Juan Zhang conceived and designed the experiments, authored or reviewed drafts of the article, and approved the final draft.
- Zichen Li conceived and designed the experiments, authored or reviewed drafts of the article, and approved the final draft.

## Data Availability

The data is available at GitHub and Zenodo:

- https://github.com/dongdong0012/DTN-demo.git

- dongdong0012. (2025). dongdong0012/DTN-demo: First release of DTN: Tracing truth: Dynamic temporal networks for multi-modal fake news detection (v1.0.0). Zenodo. https://doi.org/10.5281/zenodo.15545745.

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
