# Peer review of "Tracing truth: dynamic temporal networks for multi-modal fake news detection"

_PeerJ Computer Science, doi:10.7717/peerj-cs.2998_

## Round 0.1 · original submission · Major Revisions

Authors should address major revisions before the paper can be considered for review again.

While the work has value, reviewers present concerns about decisions made in the study, such as the selection of datasets and baseline models, which could be expanded and further justified.

**Language Note:** The review process has identified that the English language must be improved. PeerJ can provide language editing services - please contact us at [email protected] for pricing (be sure to provide your manuscript number and title). Alternatively, you should make your own arrangements to improve the language quality and provide details in your response letter. – PeerJ Staff

·

Basic reporting

The manuscript is written in generally understandable English, but it would benefit from proofreading to improve clarity and professionalism. There are several instances of grammatical or formatting issues that hinder comprehension. For example, in the introduction a sentence reads “... diverse.However, the proliferation of fake news has…”​, missing a space after the period, and “In response,This study proposes…”​where a space is missing after the comma and 'This' should be lowercase. Also, transitional words like “Firstly,this model…” should be “Firstly, this model”​. Such minor errors (missing spaces, punctuation, and capitalization) occur throughout the paper The authors should thoroughly copy-edit the text to ensure proper spacing and grammar.

The introduction is detailed and sets up the research questions well. The key contributions are even enumerated later (e.g., introducing a time similarity metric, a dynamic fusion mechanism, etc.), which is very helpful. One minor issue: the term “multi-modal” is used, but it appears the modalities considered are textual content and social/network features (there is no explicit use of image or video content in the experiments, as far as one can tell). For clarity, the authors should explicitly state what modalities are fused in DTN. If only textual and user/network data are used, calling it “multi-modal” is slightly misleading – perhaps “multi-source” (news content + social context) would be more precise, or the authors should clarify that image or other media features were not incorporated in this study.

The literature review is relevant and helps situate the DTN model, but a couple more references to prominent multimodal detection studies would strengthen it further. Additionally, there are a few formatting inconsistencies in the in-text citations that should be fixed. In several places, author names appear duplicated (e.g., “YinYin et al. (2024)” and “BianBian et al. (2020)” instead of Yin et al. and Bian et al.)​. Likewise, “XueXue et al. (2021)” is presumably a formatting error​. These seem to be typos or reference formatting issues; the authors should correct these to avoid confusion.

The figures and tables included are pertinent and sufficiently described, meeting the expectations for professional presentation.

Experimental design

The authors tackle the problem of fake news detection using a nmodel (DTN) that integrates multi-modal data with dynamic temporal network analysis. The methodology appears rigorous and leverages fine techniques (e.g., pre-trained language models, graph networks). One concern is that certain parts of the methods section are not explained with enough clarity, which could hinder reproducibility. For instance, the textual feature extraction process is confusingly described. The manuscript mentions that a fine-tuned RoBERTa model is used to obtain a [CLS] embedding for a text sequence​, and then it states “The T5 model encodes the news title and content to produce embedding vectors, subsequently normalized…”. It is unclear why both RoBERTa and T5 are needed – are these used in parallel for different pieces of text (perhaps RoBERTa for social media posts and T5 for the main news content)? The relationship between the RoBERTa-derived embedding e and the T5-derived embeddings (e_t for title, e_c for content, e_p for posts) is not clearly explained. This could be an inconsistency or an oversight in the writing. The baseline models are all out of date, the DTN should be compared with more recent baselines, such as: COOLANT, AMPLE, SAMPLE, which are introduced in recent years.

Validity of the findings

The study uses two datasets, PHEME and GossipCop, which are standard in fake news detection research. However, as the author utilzed the FakeNewsNet, why the PolitiFact, which similar to the GossipCop, is not implemented in this work? The conclusion states that “DTN surpasses all baseline methods”​, which suggests a comprehensive comparison. One suggestion is that the authors ensure they compare against strong baselines, including not only text-only models but also any prior multi-modal or graph-based models. If a particular state-of-the-art model (for example, a recent graph-based fake news detector or a multimodal network from 2022–2023) was omitted, the authors should justify why or include it.

There is some inconsistency in terminology within the findings/discussion. The authors refer to the “Temporal Dynamic Monitoring Module (DMT)” in the ablation study​, but in the methods description earlier, the same component was called the “Dynamic Temporal Analysis Module”. This inconsistency in naming could confuse readers when matching the modules to the results. The authors should standardize the naming of modules across the paper.

Additional comments

no comment

Reviewer 2 ·

Basic reporting

The manuscript is mostly clear and professionally written, but there are some awkward phrases and grammatical issues that need polishing.

Experimental design

The overall design is sound and well-structured, addressing a real challenge in fake news detection through multi-modal and temporal modeling.

Validity of the findings

The proposed model performs well on benchmark datasets and shows clear improvements through ablation. However, comparisons are missing with recent state-of-the-art models from 2023–2024. Additional evaluation under noisy conditions and for early-stage detection would strengthen the paper’s real-world value.

Additional comments

No

Cite this review as

Reviewer 3 ·

Basic reporting

This paper proposes a model DTN, Dynamic Temporal Network, for multi-modal fake news detection, which comprehensively integrates textual semantics, user behavior, propagation paths, and temporal dynamics. The model constructs a heterogeneous graph to simulate the propagation structure in social networks, designs a temporal similarity mechanism to capture the chronological relationships between nodes, and fuses multi-modal features to enhance semantic representation. To further model the propagation behavior, this paper introduces entropy-based analysis to quantify the diffusion patterns. Extensive experiments demonstrate that DTN outperforms existing methods across multiple datasets, showcasing strong detection accuracy and robustness.

Experimental design

Strengths:
(1)The proposed DTN integrates heterogeneous graph construction, temporal dynamics analysis, and multi-modal semantic fusion, showcasing notable architectural innovation. It effectively captures the propagation characteristics of fake news from multiple perspectives.
(2)By incorporating temporal modeling mechanisms, the model enhances its ability to identify fake news under various time scales.
(3)Extensive experiments not only validate the superior performance of the DTN model on two real-world datasets, PHEME and GossipCop, but also systematically evaluate the impact of learning rate and batch size on model performance. Additionally, ablation studies are conducted to demonstrate the contribution of each module to the overall effectiveness of the model.

Validity of the findings

Weakness:
(1)The use of a single Transformer encoder for global semantic modeling may not be sufficient to capture the complex interactions among multi-modal features. More specialized methods for cross-modal representation learning could be considered.
(2)Although each module is described in detail, the explanations are somewhat fragmented and overly complex.
(3) Some closely related references are missing. It is recommended that authors consider:
A Survey of Multimodal Fake News Detection: A Cross-Modal Interaction Perspective
Human cognition-based consistency inference networks for multi-modal fake news detection, TKDE 2023.
See how you read? multi-reading habits fusion reasoning for multi-modal fake news detection. AAAI 2023

Additional comments

(1)On page 9, line 308, the term MeanPool is mentioned, but it does not appear in Equation (7), which also lacks references to the Bi-RNN mentioned in the preceding paragraph. This inconsistency between the textual description and the mathematical formulation should be clarified.
(2)It is recommended to include a case study that compares the prediction outcomes of different news examples, which would enhance the interpretability and practical relevance of the proposed model.
The main architectural diagram of the model is not sufficiently clear, making it difficult for readers to intuitively understand the function of each module and how they interact. Improving the visual clarity of this diagram is strongly suggested.

Cite this review as

---

## Round 0.2 · accepted · Accept

Based on the re-review of one of the original reviewers as well as my own checks, I'm happy to recommend acceptance of this paper in its current form.

Reviewer 2 ·

Basic reporting

The authors have incorporated the suggestions in the paper in the well defined way.

Experimental design

Now, it is updated.

Validity of the findings

NO

Additional comments

NO

Cite this review as